# Effects of Different Doses of Caffeinated Coffee on Muscular Endurance, Cognitive Performance, and Cardiac Autonomic Modulation in Caffeine Naive Female Athletes

**DOI:** 10.3390/nu13010002

**Published:** 2020-12-22

**Authors:** Raci Karayigit, Alireza Naderi, Firat Akca, Carlos Janssen Gomes da Cruz, Amir Sarshin, Burak Caglar Yasli, Gulfem Ersoz, Mojtaba Kaviani

**Affiliations:** 1Faculty of Sport Sciences, Ankara University, Gölbaşı, Ankara 06830, Turkey; rkarayigit@ankara.edu.tr (R.K.); fakca@ankara.edu.tr (F.A.); gersoz@ankara.edu.tr (G.E.); 2Department of Sport Physiology, Boroujerd Branch, Islamic Azad University, Boroujerd 6915136111, Iran; Naderi_a@yahoo.com; 3Laboratory of Exercise Physiology, Faculty of Physical Education, University of Brasilia, Brasilia 70910-900, Brazil; janssengomes@gmail.com; 4Department of Exercise Physiology, Faculty of Physical Education and Sport Sciences, Karaj Branch 3149968111, Iran; amsarshin@gmail.com; 5Department of Physical Education and Sports, Iğdır University, Igdir 76000, Turkey; burakcaglar90@gmail.com; 6Faculty of Pure & Applied Science, School of Nutrition and Dietetics, Acadia University, Wolfville, NS B4P 2R6, Canada

**Keywords:** ergogenic aid: female athletes, strength, coffee

## Abstract

Caffeine is widely consumed among elite athletes for its well-known ergogenic properties, and its ability to increase exercise performance. However, studies to date have predominantly focused on the anhydrous form of caffeine in male participants. The aim of the study was to investigate the effect of caffeinated coffee ingestion on lower-upper body muscular endurance, cognitive performance, and heart rate variability (HRV) in female athletes. A total of 17 participants (mean ± standard deviation (SD): age = 23 ± 2 years, body mass = 64 ± 4 kg, height = 168 ± 3 cm) in a randomized cross-over design completed three testing sessions, following the ingestion of 3 mg/kg/bm of caffeine (3COF), 6 mg/kg/bm of caffeine (6COF) provided from coffee or decaffeinated coffee (PLA) in 600 mL of hot water. The testing results included: (1) repetition number for muscular endurance performance; (2): reaction time and response accuracy for cognitive performance; (3): HRV parameters, such as standard deviation of normal-to-normal (NN) intervals (SDNN), standard deviation of successive differences (SDSD), root mean square of successive differences (RMSSD), total power (TP), the ratio of low- and high-frequency powers (LF/HF), high-frequency power (HF), normalized HF (HFnu), low-frequency power (LF), and normalized LF (LFnu). A one-way repeated measures ANOVA revealed that 3COF (*p* = 0.024) and 6COF (*p* = 0.036) improved lower body muscular endurance in the first set as well as cognitive performance (*p* = 0.025, *p* = 0.035 in the post-test, respectively) compared to PLA. However, no differences were detected between trials for upper body muscular endurance (*p* = 0.07). Lastly, all HRV parameters did not change between trials (*p* > 0.05). In conclusion, ingesting caffeinated coffee improved lower body muscular endurance and cognitive performance, while not adversely affecting cardiac autonomic function.

## 1. Introduction

Coffee contains caffeine, and is the second most consumed beverage behind water, with more than 2 billion (estimated) cups consumed globally per day. It is commonly consumed by athletes to improve physical and cognitive performance [1]. It has been well established that 3–6 mg per kilogram of body mass (mg/kg/bm) of caffeine usage, 60 min prior to exercise, improves strength and aerobic-muscular endurance performance via the most probable mechanisms, with adenosine A_1_ and A_2A_ receptor antagonism, and NA^+^/K^+^ ATPase pump activation [2,3]. In addition, the hypoalgesic effect of caffeine may decrease muscle pain perception during high-intensity resistance exercise that induces pain through blocking central and peripheral adenosine receptors that influence pain signaling [4]. Considering the improvements in reaction time, cognition, and mood after caffeine consumption, Astorino ve Roberson [3] suggested that the ergogenic mechanisms of caffeine are likely to be multifactorial and have the potential to enhance performance. However, doses above 500 mg may cause the opposite reaction, with performance decrements, anxiety, and tension [2,3].

Despite the initial controversy on whether biologically active compounds (ferulic acid, chlorogenic acid, caffeic acid, etc.) found in coffee blunts the ergogenic effects of caffeine [5,6,7], the use of coffee as a form of caffeine administration may have comparable effects to anhydrous caffeine on exercise performance [8,9,10,11]. Trexler et al. [11], in their between-group design study, reported anhydrous caffeine is not superior to caffeinated coffee for sprint and muscular endurance performance and pointed out the interindividual variation in responses to caffeine intake in the anhydrous caffeine group may be related to large standard deviations in self-reported habitual daily caffeine consumption. The standardization of individual habituation may make the ergogenic or ergolytic effect of caffeine more detectable [12]. Similarly, Richardson and Clarke reported that number of squat repetitions performed under decaffeinated + anhydrous caffeine conditions were significantly greater than those performed under decaffeinated coffee, anhydrous caffeine, and placebo conditions. Further, bench press performance after caffeinated coffee ingestion was not enhanced with similar magnitude to anhydrous caffeine [10]. Although the multiform effects of caffeine on strength and endurance of lower and upper body has been reported by previous studies [13,14,15,16,17], further investigations as to whether improvement in lower body endurance with caffeine intake persists over multiple sets in a dose dependent manner was suggested [10].

Excess anhydrous caffeine intake can lead to gastrointestinal problems, tachycardia, sinus, tingling sensations, and negative health effects with long-term use [2], and dose–response studies have increased recently [18,19,20,21,22]. However, most of them suggested between 3–6 mg/kg/bm classically; the optimal dose required to have ergogenic effects may differ based on sex [21], muscle group size [18], and habitual caffeine consumption [22]. Evident differences between sexes in terms of body size, lean body mass, and hormonal functioning may affect the results for the same dosage of caffeine [23]. Given that female participants accounted for only 13% of the total number of participants included in the studies investigating the ergogenic effect of caffeine [23], it would be difficult to generalize the recommendations. This was confirmed by Sabblah et al., showing 5 mg/kg/bm of anhydrous caffeine increased 1 repetition maximum (RM) strength performance in both males and females, but for muscular endurance, tendency towards improvement appeared in males only [21]. In another study, by Arazi et al., 5 mg/kg/bm, but not 2 mg/kg/bm of anhydrous caffeine, decreased pain perception during muscular endurance tests in female karate athletes [20]. Inconsistent results between males and females can also be related to oral contraceptive use that may decrease the caffeine clearance, which was considered by only a few studies [23]. Further, both 3 and 6 mg/kg/bm of anhydrous caffeine doses significantly improved lower body muscle strength, with no effect on upper body reported by Tallis and Yavuz [24]. Moreover, only 6 mg/kg/bm of caffeine increased peak force during repeated contractions, inferring optimal doses to improve muscle performance may be related to specific muscle mass. However, none of the above studies were conducted with coffee forms and on female athletes. In addition, 6 mg/kg/bm of caffeine intake using coffee consumption equates to 2 cups of strong-tasting coffee, which may irritate the gastrointestinal tract. Coffee consumption, especially in the morning hours, may not be appealing to athletes because of time commitment to ingest strong and hot beverages [10]. All of these studies in the literature point out that the effects of different doses of caffeinated coffee on muscular endurance of the lower and upper body, over multiple sets, should be examined in females.

Diurnal variation in muscles and cognitive performance oscillate higher at midday, and lower during the early morning hours [25]. Additionally, in congested match fixture, athletes can have training very early in the morning, unsurprisingly, with declining muscle performance that affect long-term training adaptations. To overcome this, higher beneficial effects of caffeine in the morning may be considered. A related study by Mora-Rodriguez et al. reported that 6 mg/kg/bm of anhydrous caffeine counteracted the decline in muscle performance observed in the morning, but had little effect on neuromuscular performance, and increased the rate of negative side-effects reported in the evening [26]. However, effects of lower doses and coffee form on muscle and cognitive performance in the early morning were not investigated. Further, beneficial effects of caffeine on cognitive function including increased arousal, reaction time, and vigilance in sports settings has been well established [27,28]. There is a limited number of studies that investigate caffeine intake on cognitive performance in female athletes. Only Ali et al. examined female athletes’ responses to 6 mg/kg/bm of anhydrous caffeine, showing non-significant but positive trends (*p* = 0.072) in cognitive function [29]. Although the positive effect of 3 mg/kg/bm of anhydrous caffeine on the velocity of half-squat exercise [30] and peak aerobic cycling power [31] were reported to be similar in all three phases of the menstrual cycle, Kumar et al. demonstrated cognitive performance declines in the high progesterone, luteal phase of the menstrual cycle [32]. Muscle performance is defined by the characteristics of a complex network of mental and physical elements, investigating caffeine’s effect on cognitive performance will help us better understand its ergogenic properties. Furthermore, despite the documented ergogenic effect of caffeine on exercise performance, knowledge of its effect on cardiac recovery is somewhat limited [33,34,35]. The current study is the first to investigate the effects of various doses of caffeinated coffee, which may minimize overdose risk with its relatively low caffeine/volume ratio. Therefore, the aim of this study was to investigate the effects of different doses of caffeinated coffee ingestion on muscular endurance, cognitive performance, and heart rate variability (HRV) in female athletes.

## 2. Materials and Methods

### 2.1. Participants

Seventeen (rugby: 10, handball: 4, and soccer: 3) healthy, non-smoking, resistance-trained female team sport athletes volunteered to participate in this study (age, 23 ± 2 years; height, 168 ± 3; weight, 64 ± 4 kg; resistance training experience, 4 ± 1 years; caffeine intake, 15 ± 4 mg/day; squat 1 repetition maximum (1 RM), 92.0 ± 7.1 kg; barbell bench press 1 RM, 64.5 ± 8.3 kg). Rugby players were competing in the Turkish College Rugby Super League and four of them were members of the Turkey national rugby team. Handball and soccer players were participating in teams, competing at the second and third professional league, at the national level of their branch, respectively. Participants had at least 3 years of resistance training experience, three times per week for the previous year, and included full squat and bench press exercises in their training routines. All subjects stopped taking substances that could have affected the muscular and cognitive performance (i.e., creatine, anabolic steroids, and oral contraceptives) for at least 3 months before the commencement of the study. Habitual caffeine consumption was assessed using an adapted version of the caffeine consumption questionnaire proposed by Buhler et al. [36], under the supervision of a qualified nutritionist. Individual amounts of caffeine containing food during a day, week, and month was assessed using household measures. Previous studies and the Turkish food composition database were searched and used for database construction [12,36,37]. All participants were naive consumers (<25 mg/day), according to the classification recently proposed by Filip et al. [38] to standardize the categorization of athletes, and in turn, reduce the likelihood of discrepancies in qualification of daily caffeine intake between studies. Participant inclusion criteria were as follows: (a) free from neuromuscular and musculoskeletal disorders, aged 18–30 years, (b) able to perform successful back squat and bench press exercises with load corresponding to 125% and 100% of their current body mass, respectively (c) non-smoker, and (d) naïve caffeine consumer [38]. Written informed consent outlining the purpose, procedures, and protocol risks were obtained from all participants; the study procedures followed the principles outlined in the Declaration of Helsinki and were approved by Ankara University, Human Research Ethics Committee (16-1015-17).

### 2.2. Study Design

A double-blind, randomized, counter-balanced, crossover research design was used. Each participant attended the laboratory on five occasions, separated by 48–72 h to allow recovery. The arrangements concerning session order, timing, and blinding was conducted by a researcher who was not involved in data collection and analyses. The first two sessions were used for familiarization and the following three sessions were employed to complete the test protocol with the ingestion of 6 mg/kg/bm of caffeine provided from coffee (6COF), 3 mg/kg/bm of caffeine provided from coffee (3COF), and decaffeinated coffee (PLA) before the exercise protocol. Body composition (Jawon Segmental Avis 333 Plus, Korea) and 1 RM tests were measured in familiarization sessions. Strength (1 RM) tests were completed according to protocols outlined by Baechle and Earle [39] and Richardson and Clarke [10]. In order to measure muscular endurance performance in the subsequent sessions, a weight equivalent to 40% of 1 RM was calculated and recorded for each exercise, proportionally to the higher 1 RM value established from two familiarization sessions. Participants were introduced to the felt arousal (FA) scale [40] and valid muscle pain perception scale (MP) [41] used in previous studies [4,10] to monitor arousal and muscle pain throughout the protocol. MP scale described by Cook et al. [41], which ranged from 0 to 10, with 0 representing “no pain at all” and 10 “extremely intense pain” was used. Participants also practiced with the cognitive function (CF) test on the computer until achievement of consistent scores. Upon arrival at the testing site, participants undertook HRV and CF measurements followed by coffee ingestion, and were allowed 1-h passive rest in the seated position, and to consume 500 mL of water once the coffee mug was emptied. Then, HRV and CF tests were repeated. Following these tests, participants completed 5 min jogging on a treadmill and standardized static-dynamic stretching. Static stretching targeted the quadriceps and hamstrings, while participants laid in a supine or on their sides and arms, with a 90 degree bend in the arm, pushed straight across the front of the body, or pushed back and down the back of the body behind the head, which was executed for 30 s with a 30 s rest, followed dynamically by two stretches while standing—one for the upper body and another for the lower body—such as moving one arm forward or backward in a diagonal plane, and one leg backward and then forward in a sagittal plane (which took about 30 s for each exercise) [42]. Then participants started the 3 sets of 40% of 1 RM repetitions to failure squat protocol, with a 2-min rest between sets. The same protocol was replicated for the bench press after a 5 min passive rest period. Immediately after muscular endurance tests, HRV and CF were measured, respectively. Heart rate (HR) (Polar Team 2 telemetric system, Finland), capillary lactate (LA) (Lactate Scout, USA), and glucose (GLU) (Accutrend Plus, Roche Diagnostics, Mannheim, Germany) from a fingertip, FA and MP were measured at different time points throughout the test protocol. Figure 1 displayed a schematic of the experimental protocol. Given that participants in this study typically skipped the breakfast due to the congested match and training schedule, all test sessions took place in the early morning hours (6:30–7:30 a.m.), following an overnight fast to simulate their real-life settings. On the day preceding the first session, participants recorded their 24-h dietary intake and then photocopied and handed back to replicate this diet before subsequent sessions. Participants were instructed to avoid caffeine, alcohol ingestion, and strenuous exercise for 24 h prior to the testing sessions. Adherence to diet and avoidance procedures were checked verbally before each session. Although increase in the resistance exercise performance was reported to show a similar magnitude in all three phases of the menstrual cycle [30], cognitive performance [32] and pain perception tolerance appeared to decline [43] in the luteal phase. Therefore, all sessions were performed during the luteal phase of the participants’ menstrual cycle, of which was pursued for 4 months before the onset of the experiment via a mobile application (menstruation calendar—period tracking, TR). Luteal phase was determined by individual declaration of the 20th day from the first day of the menstrual cycle (beginning of menses) [31].

### 2.3. Strength (1 RM) and Muscular Endurance Test Protocol

Participants warmed up with 20 kg weights for 10 repetitions, then rested passively for 1 min, followed by a further 3–5 repetitions, with an added 20% and 10% more weight for the back squat and barbell bench press, respectively. A 2-min passive rest was allowed and then participants chose a weight near 1 RM to lift for 2–3 repetitions. The participant rested for 3 min, the weight was increased by 10–20% for squat and 5–10% for bench press, and the participant performed the first 1 RM attempt. If the load was successfully lifted, following a 3-min rest, the weight was increased again by 10–20% for squat and 5–10% for bench press, and 1 RM was attempted. If the lift was unsuccessful, the weight was decreased by 5–10% for squat and 2.5–5% for bench press for another 1 RM attempt after a 3-min rest. Furthermore, 1 RM was identified in 3–5 steps [10,39]. Following a 1 RM establishment, participants performed three sets of repetitions to failure with the load of 40% of 1 RM for the squat and bench press. Participant performance of the back squat on a smith machine (Esjim, Eskişehir, Turkey) and the bench press exercises, on a rack with safety bars and Olympic plates (Esjim, Eskişehir, Turkey), was checked by a certified personal trainer to standardize lifting technique, and proper feedback was given when needed. The foot placement and wide bar grip position [44] of each participant during squat and bench press exercises were recorded and replicated for subsequent main sessions to ensure that same stance was used in all lifting. During both the 1 RM and repetitions to failure protocols for squat and bench press exercises, repetitions were performed at a speed of 2 s for both eccentric and concentric phases, due to the strength and strength endurance tests, was suggested to perform at the same tempo [45,46], which was controlled using a metronome (mobile application). During the squat, participants were instructed to descend until their thighs were parallel with the ground, and ascended after touching the tightrope placed parallelly at this squad depth with their buttocks. Every repetition in the bench press was lowered to the chest and then raised until elbows were fully extended to standardize the range of motion. Total repetition number during each set was recorded as an index of muscular endurance performance [47]. Three criteria were appointed to complete the repetition to failure protocol; (1) the participant voluntarily terminated the repetition; (2) the participant was unable to lift synchronously with the metronome for three consecutive repetitions; (3) the participant was unable to perform an additional repetition with proper technique and posture.

### 2.4. Caffeinated Coffee Protocol

Decaffeinated (as a placebo) and caffeinated coffee was prepared using instant coffee (Nescafé Gold, Nestlé, Turkey). Caffeine content of the caffeinated coffee was calculated according to the measurement of a reference amount taken from the same batch by Ankara Food Control Laboratory Directorate, reporting that 100 gr caffeinated coffee contains 36 mg of caffeine, so participants ingested 0.16 gr/kg/bm caffeinated coffee to intake 6 mg/kg/bm caffeine, 0.16 gr/kg/bm decaffeinated coffee as placebo, 0.08 gr/kg/bm caffeinated + 0.08 gr/kg/bm decaffeinated (totally 0.16 gr/kg/bm) coffee to consume 3 mg/kg caffeine. Nescafe decaffeinated coffee was reported to provide 0.17 mg/kg/bm caffeine, meaning a very low dose to be ergogenic [10]. To detect caffeine’s erogenicity in a dose dependent manner, and exclude other biologically active compound effects on muscle performance, participants ingested equal volume (0.16 gr/kg/bm) of coffee with different amounts of caffeine content in each session. Participants had 10 min to consume the coffee dissolved in 600 mL of hot water (approximately 70 °C) served in a mug [10]. Sixty minutes of time was given between coffee intake and exercise tests to allow complete caffeine absorption.

### 2.5. Heart Rate Variability

The HRV measurements (approximately 5 min) at baseline, 60 min after coffee ingestion and post-test, were obtained with an Omega Wave 800 (OW, Portland, OR, USA) device [48], while participants were in the supine position. First, HRV measurement at baseline was taken following a 15 min passive supine position rest. Three of the seven electrodes used during measurements were thoracic Wilson electrodes and four of them were tarsale limb electrodes. Participants were asked to remain silent and still during the measurements while maintaining their routine respiratory rate. The parameters obtained from HRV measurements were standard deviation of normal-to-normal (NN) intervals (SDNN), standard deviation of successive differences (SDSD), root mean square of successive differences (RMSSD), which indicates parasympathetic activity, total power (TP) as an indication of variations between NN intervals, the ratio of low- and high-frequency powers (LF/HF), which represents sympathovagal balance, high-frequency power (HF), which denotes vagal activity, normalized HF (HFnu), low-frequency power (LF), which depicts combination of sympathetic and parasympathetic activity, and normalized LF (LFnu). All variables were automatically calculated by software (Omega Wave Sport Tech, Portland, OR, USA).

### 2.6. Cognitive Function

The modified arrow version of flanker task [28,49] was used to measure CF and run on a Dell Computer using Inquisit Lab 5.0 (Milliseconds) software. On each trial, a central yellow fixation star was replaced by five black arrows in a line focally presented for 200 ms on a white background with a response window of 1000 ms. Participants were instructed to respond as quickly and accurately as possible to the direction of the middle target arrow, ignoring two flanker arrows on left and right sides, by pressing corresponding response buttons with their left or right index fingers. Half of the trials were congruent (< < < < < or > > > > >), whereas the other half were incongruent (e.g., < < > < < or > > < > >). Trial order was random for each participant and the inter-stimulus interval varied from 1100 to 1500 ms. After completing 20 practice trials, participants were administered 100 trials by wearing earplugs. Total duration of the flanker task was approximately 3 min. To measure cognitive performance, mean response accuracy (%) and response times (ms) were collected [49].

### 2.7. Statistical Analysis

All data were analyzed using the IBM SPSS statistics for Windows, version 22.0 (IBM Corp., Armonk, NY, USA). Normality of the data was verified using Shapiro–Wilk test, statistical significance of 1 RM measurements were examined using Student’s *t*-test for paired samples. Data were analyzed using two-way analysis of variance (ANOVA) for repeated measures to examine main effects for (1) condition, (2) time or set, and (3) condition x time or set interaction. Sphericity was analyzed by Mauchly’s test of sphericity followed by the Greenhouse–Geisser adjustment where required. If any differences were identified, post hoc Bonferroni adjustment was conducted. Statistical significance was set at *p* < 0.05 and data were presented as mean ± SD. Intraclass correlation coefficients (ICC) were computed to assess the consistency or test–retest reliability of the three trials with conditions; 95% confidence interval (CI) and the effect sizes were calculated using partial eta squared (η^2^), defined as trivial (<0.10), moderate (0.25–0.39), or large (≥0.40) [50].

## 3. Results

### 3.1. Strength (1 RM) Performance during Familiarization Sessions

There was no significant difference between first and second familiarization session in 1 RM performance for squat (first: 91.2 ± 7.4, (83.7–94.9), second: 92.0 ± 7.1 (88.3–95.7)) (*p* = 0.26), and bench press (first: 63.5 ± 6.7 (60.0–66.9), second: 64.5 ± 8.3 (60.2–68.8)) (*p* = 0.23) exercises, respectively. All variables showed high ICC for squat (0.95) and bench press (0.94) can be seen in Table 1.

### 3.2. Muscular Endurance Performance

In two-way ANOVA, condition x set interaction (*p* = 0.01, η^2^ = 0.30) and main effect for set (*p* = 0.01, η^2^ = 0.94) were detected, but there was no main effect for condition (*p* = 0.11, η^2^ = 0.13) in squat. One-way ANOVA showed significant differences in squat muscular endurance performance between conditions in the first set (6COF: 33.9 ± 6.0, 3COF: 33.4 ± 5.2, PLA: 31.2 ± 5.1; *p* = 0.01; η^2^ = 0.31). Post-hoc analysis revealed that for caffeinated coffee conditions, participants performed significantly more repetitions in 3COF (*p* = 0.03, 95% CI: 1.2–4.2) and 6COF (*p* = 0.02, 95% CI = 0.3–5.0) compared to PLA in the first set. There was no significant difference between 3COF and 6COF conditions in the first set (*p* = 0.93, 95% CI = 0.8–1.8). Further, no significant difference was detected in the second and third sets between all conditions (*p* > 0.05). No main effect for conditions in the bench press exercise was detected (*p* = 0.80, η^2^ = 0.01). There was a significant main effect for set (*p* = 0.01, η^2^ = 0.91). In addition, two-way ANOVA showed a trend for condition x set interaction (*p* = 0.07, η^2^ = 0.12). Despite non significance, post-hoc analysis pointed a trend in the first set for 6COF condition compared to PLA (*p* = 0.06, 95% CI = 0.04–1.5). Participants performed more repetitions (3.1%) in the first set of the 6COF condition, in comparison to PLA. There was also no significant difference between 6COF and 3COF (*p* = 0.47, 95% CI = 0.5–1.9). In addition, there were no significant difference or trend in the second and third sets between all conditions (Figure 2). ICC, for squat, was 0.94 (95% CI = 0.86–0.97), 0.97 (95% CI = 0.95–0.99), 0.96 (95% CI = 0.93–0.98) for the first, second, and third sets, respectively. ICC results from bench press was 0.97 (95% CI = 0.95–0.99), 0.95 (95% CI = 0.90–0.98), 0.94 (95% CI = 0.87–0.97) for the first, second, and third sets, respectively.

### 3.3. HRV

Two-way ANOVA showed the main effect for time in all HRV parameters, including SDNN (*p* = 0.01, η^2^ = 0.74), SDSD (*p* = 0.01, η^2^ = 0.67), RMSSD (*p* = 0.01, η^2^ = 0.65), TP (*p* = 0.01, η^2^ = 0.83), LF (*p* = 0.01, η^2^ = 0.73), HF (*p* = 0.01, η^2^ = 0.73), LFnu (*p* = 0.01, η^2^ = 0.67), HFnu (*p* = 0.01, η^2^ = 0.74), LF/HF (*p* = 0.01, η^2^ = 0.92). However, no main effect for condition in SDNN (*p* = 0.92, η^2^ = 0.01), SDSD (*p* = 0.65, η^2^ = 0.02), RMSSD (*p* = 0.67, η^2^ = 0.02), TP (*p* = 0.93, η^2^ = 0.01), LF (*p* = 0.57, η^2^ = 0.03), HF (*p* = 0.90, η^2^ = 0.01), LFnu (*p* = 0.28, η^2^ = 0.07), HFnu (*p* = 0.85, η^2^ = 0.01), LF/HF (*p* = 0.59, η^2^ = 0.03) and condition x time interaction in SDNN (*p* = 0.63, η^2^ = 0.03), SDSD (*p* = 0.82, η^2^ = 0.02), RMSSD (*p* = 0.84, η^2^ = 0.02), TP (*p* = 0.88, η^2^ = 0.01), LF (*p* = 0.92, η^2^ = 0.01), HF (*p* = 0.85, η^2^ = 0.02), LFnu (*p* = 0.16, η^2^ = 0.09), HFnu (*p* = 0.47, η^2^ = 0.05), LF/HF (*p* = 0.88, η^2^ = 0.01) were detected (Table 2).

### 3.4. Cognitive Performance

Results from the flanker task for response accuracy in two-way ANOVA showed no significant condition x time interaction (*p* = 0.92, η^2^ = 0.01), or main effects for condition (*p* = 0.36, η^2^ = 0.06), or time (*p* = 0.41, η^2^ = 0.05), for the congruent condition. Response accuracy in the incongruent condition also showed no significant condition x time interaction (*p* = 0.11, η^2^ = 0.11), or main effects for treatment (*p* = 0.42, η^2^ = 0.05), but significant main effect for time was detected (*p* = 0.04, η^2^ = 0.17) in two-way ANOVA. Regarding reaction time in the congruent condition, there were significant main effects for condition (*p* = 0.01, η^2^ = 0.21), time (*p* = 0.01, η^2^ = 0.42), and condition x time interaction (*p* = 0.01, η^2^ = 0.21). Post-hoc pairwise comparison for the main effect for condition showed 6COF significantly faster reaction time than PLA (*p* = 0.03), but no differences were detected between 3COF and PLA (*p* = 0.56), and between 6COF and 3COF (*p* = 0.34). Further, one-way ANOVA applied for significant condition x time interaction showed only 6COF has significantly faster reaction time than PLA at the post test (*p* = 0.01). In addition, Bonferroni post-hoc pairwise comparison showed a trend with faster reaction time at the post test for 3COF condition compared to PLA (*p* = 0.10). Similar results appeared for reaction time in the incongruent condition, there were significant main effects for condition (*p* = 0.01, η^2^ = 0.32), time (*p* = 0.01, η^2^ = 0.64), and condition x time interaction (*p* = 0.02, η^2^ = 0.15) in two-way ANOVA. Post-hoc pairwise comparison for the main effect for condition showed significantly faster reaction time for 6COF than PLA (*p* = 0.01) and for 3COF than PLA (*p* = 0.02), and no difference was found between 6COF and 3COF (*p* = 0.99). Further, one-way ANOVA showed 6COF has significantly faster reaction time than PLA (*p* = 0.01) and a trend was found for 3COF compared to PLA (*p* = 0.06) at the post coffee ingestion. Further, 6COF (*p* = 0.02) and 3COF (*p* = 0.03) were significantly faster compared to PLA at the post-test (Table 3).

### 3.5. Lactate, Glucose, Heart Rate, Pain Perception, Arousal

Two-way ANOVA analysis showed that there was a significant condition x time interaction (*p* = 0.01, η^2^ = 0.22) for lactate, and there were significant main effects for condition (*p* = 0.01, η^2^ = 0.51) and time (*p* = 0.01, η^2^ = 0.93). Post-hoc pairwise comparison for the main effect for condition showed both 6COF (*p* = 0.01) and 3COF (*p* = 0.01) were significantly higher compared to PLA. Further, one-way ANOVA showed a significant difference only for 6COF at the post-squat compared to PLA (*p* = 0.03). In addition, at the post-bench press, both 6COF (*p* = 0.01) and 3COF (p= 0.01) were significantly higher compared to PLA. There was no significant main effects for condition (*p* = 0.11, η^2^ = 0.12, time (*p* = 0.41, η^2^ = 0.05) and condition x time (*p* = 0.39, η^2^ = 0.06) interaction for glucose. Similarly, no significant main effects for condition (*p* = 0.17, η^2^ = 0.10) and condition x time interaction (*p* = 0.67, η^2^ = 0.04) were found for heart rate, but there was a significant main effect for time (*p* = 0.01, η^2^ = 0.99). 

Pain perception showed no significant main effect for condition (*p* = 0.48, η^2^ = 0.04), while main effect for time (*p* = 0.01, η^2^ = 0.35) and condition x time (*p* = 0.01, η^2^ = 0.24) interaction were found significant. One-Way ANOVA showed that pain perception was only significantly lower in 6COF condition at post-squat time point compared to PLA (*p* = 0.03). However, no significant differences were found between the conditions at the post-bench press time point. 

Two-way ANOVA results of arousal showed that There was significant main effects for condition (*p* = 0.01, η^2^ = 0.32), time (*p* = 0.01, η^2^ = 0.83), and condition x time interaction (*p* = 0.01, η^2^ = 0.46). Post-hoc pairwise comparison showed both 6COF (*p* = 0.01) and 3COF (*p* = 0.01) were significantly higher compared to PLA. Lastly, there was no difference between any conditions prior to coffee ingestion; however, 6COF (*p* = 0.01) and 3COF (*p* = 0.01) were significantly higher at post-coffee ingestion compared to PLA (Table 4).

## 4. Discussion

To the best of our knowledge, the current study is the first to analyze the effect of various doses of caffeinated coffee on lower–upper body muscular endurance and cognitive performance and HRV in caffeine naive female athletes in the early morning. The main finding was that the ingestion of both 3COF and 6COF increased lower body muscular endurance and cognitive performance by increasing arousal, with no harmful effect on cardiac autonomic function. The difference between 3COF and 6COF was only found in muscle pain perception while 6COF had a significant effect during lower body resistance exercise, but 3COF had none. Although 6COF improves (3.1%) upper body muscular endurance performance in the first set, there was no difference between 3COF and PLA. The current results may have implications for athletes seeking to improve muscular endurance and cognitive performance with no adverse autonomic effect by pre-exercise caffeinated coffee ingestion.

In the current study, it was determined that both 3 and 6 mg/kg/bm caffeinated coffee doses increased lower body muscular endurance performance and the lack of effect on upper body performance is parallel to previous studies [10,13], and is in contrast to the studies that have reported that meaningful effects of caffeine is not related to muscle group location and enhances upper body muscular performance [4,14]. It is also contrary to a study by Duncan et al. [4], demonstrating that 5 mg/kg/bm anhydrous caffeine diluted in water increases repetition to failure performance (with the load of 60% of 1 RM), irrespective of muscle group location, beginning the test protocol with the upper body muscle group. Grgic et al. [51], in their review regarding caffeine ingestion and resistance exercise performance, reported caffeine’s effectiveness on muscular endurance was lessened as fatigue developed, thus, reducing motor unit recruitment and force production. Supportively, it was stressed by the same research group that bench press repetition to failure performance (with the load of 60% of 1 RM) did not increase with caffeine ingestion when the bench press test was the very last in the experimental protocol, therefore, the accumulated fatigue may affect participant performance, and asserted results might have been different if the bench press had been assessed at the beginning of the test protocol [15]. Although the results of the current study, and previous [10,11] results seem to ratify this assertion, no significant effect of 6 mg/kg/bm of caffeine intake on both leg and bench press repetition to failure performance (with the load of 60% of 1 RM) was reported, even when bench press preceded leg press exercise [52]. However, only one of these studies employed a design with counterbalanced order of bench press and leg press exercises when examining the effect of caffeine on upper and lower body repetition to failure performance, reported significant improvement in second and third sets of leg press, but not in bench press [53]. Although, a meta-analysis by Warren et al. [17] reported that magnitude of caffeine erogenicity may be 4–6 times greater in lower and large (knee extensors specifically) muscle groups compared to upper and small muscle groups, such as arm muscles, rationalized with various neural activation levels during maximal voluntary contractions (85–95% for knee extensors and 90–95% (minimal/no room for improvements) for other muscle groups), and promotion of greater muscle recruitment centrally, subgroup analyses pointed that muscle group locations were significant factors for strength, but not muscular endurance performance. Additionally, various responses across the upper and lower body may partially be explained with a non-equivocal number of adenosine receptors on muscle groups depending on its size [16]. Contrasting results can largely be attributed to methodological variables of studies including caffeine dose–form, resistance exercise load intensity, heterogeneous daily caffeine intake, training status, genotype, and habituation of the participants.

In the current research, attempts were made to standardize daily caffeine consumption by recruiting very low habitual caffeine users, and training status of the participants selected to detect caffeine’s subtle ergogenic effect more clearly as suggested previously [13]. Despite the equivocal findings on this topic, efficiency of caffeine can be related to training status of participants, which trained athletes can produce maximal efforts with greater motivation and muscle mass (more adenosine receptor concentration and act directly at the muscle via increased NA^+^/K^+^ pump activation and Ca^2+^ release from sarcoplasmic reticulum) [17], and have higher pain tolerance [24] compared to untrained. Furthermore, resistance-trained participants may be more sensitive to muscular endurance responses (average 4–5 more repetitions) after caffeine ingestion, especially by overcoming psychological and muscular stress through the end of low intensity (40% of 1 RM) open-end muscular endurance tests by not allowing great variations in performance, keeping it stable over consecutive test sessions to detect subtle differences, and increase statistical power compared to untrained [10]. It may also be useful to investigate the responses of participants to caffeine with different training status—equivocal findings on this topic need to be further questioned. Heart rate, lactate, and muscle pain perception values (around 170 bpm, 8 mmol/dL, and 8 points, respectively) in the current study showed that participants performing maximally and repeatability-consistency in each condition were high (ICC ranged between 0.94 and 0.97 during muscular endurance tests). Furthermore, improvement in repetition numbers (95% CI for caffeinated coffee conditions were around 0.3–5.0) is parallel to a meta-analysis concerning acute caffeine intake and isotonic muscular endurance [54].

A wide range of load intensity (range between 40 and 80% of 1 RM) was used while testing muscular endurance performance with caffeine intake in the literature [4,11,13,21]. One may speculate that, as the load intensity increases, benefits of caffeine intake seemed to lessen in a dose–dependent manner, and vice versa. Pallares et al. reported the ergogenic dose of caffeine required to increase neuromuscular performance during all-out contraction, depending on the magnitude of load with 3 mg/kg of caffeine, was enough to increase muscular performance against low loads, whereas a higher dose (9 mg/kg/bm) was necessary against higher loads [18]. It was previously reported that slow-twitch muscle fibers may be more sensitive to caffeine ingestion than fast-twitch fibers [55]. It can be speculated that caffeine may facilitate the release of calcium; in turn, exercise performance, selectively by depending on a contraction type (slow–fast twitch) of muscle, may explain different results between the current and previous studies measuring muscular endurance performance with various load intensity. However, while a lighter load (30% maximal voluntary contraction (MVC)) time to task failure performance in isokinetic dynamometer was better in caffeine (+45 s) compared to placebo in females, it was not statistically significant with lighter and heavier (70% MVC) loads [56]. Some potential limiting factors, including use of very low dose (1.5 mg/kg/bm) of caffeine, recruiting physically active but not resistance-trained individuals and nonhomogeneous daily caffeine intake (100–300 mg/day), might have influenced the results of that study. Thus, further investigations is still required to examine the caffeine responses to muscular endurance test, representing real life settings with light (40–50% of 1 RM) and heavy (70–80% of 1 RM) load intensities. A clear majority of studies did not control movement tempo of resistance exercise during muscular endurance tests that may affect results, particularly in the later stages, by destabilizing timing between eccentric and concentric phases, and may be another consideration for future research. Thinking in reverse, reporting of both 3 and 6 mg/kg/bm of caffeine to increase mean movement velocity, and likely repetition number, during 30% of 1 RM bench press [19], may explain alleged benefits of caffeine on upper body performance in studies that did not control movement tempo, and may generate a situation in favor of caffeine. Different movement velocities have unique physiological variables, and this may be affected by caffeine ingestion; it is required to investigate caffeine’s effect on resistance exercise performance with various repetition durations (2 s vs. 4 s) as well.

Moreover, 3 mg/kg/bm caffeinated coffee provided the same magnitude of performance enhancing effect as 6 mg/kg/bm in this study. This result was similar to the results of Grgic et al. [57], reporting that 6, 4, and as little as 2 mg/kg/bm caffeine may enhance lower body muscular endurance performance (back squat 60% of 1 RM) while there was no significant benefit of caffeine intake on upper body muscular performance. These results should be carefully interpreted as 23 participants with heterogenous daily caffeine intake performed tests in the morning, and 5 in the evening, whilst no individual responses were reported in the abovementioned study. Caffeine demonstrated to exert its ergogenic effect depending on the time of day [26]. It is plausible that the participants’ chronotype (expression of circadian rhythmicity), which likely influences psychophysiological and cognitive responses to physical exercise with “morning larks” might have augmented caffeine erogenicity compared to “night owls” or vice versa [25]. To date, no study has examined the relationship between the ergogenic effect of caffeine and chronotype, and it would be a great topic for future studies to investigate interaction between caffeine and participants’ chronotype on exercise performance, particularly in a dose-dependent manner, to prescribe individual caffeine consumption strategies, depending on time of the day.

Due to regular caffeine ingestion prior to matches and training, athletes may develop tolerance to caffeine. Beneficial responses to low doses of caffeine in the current study, due to the caffeine naive participants, cannot be generalized to the athletes who routinely ingest caffeine. Supportively, Wilk et al. [22] reported that caffeine doses between 9 and 11 mg/kg/bm did not improve resistance exercise performance in athletes habituated to caffeine, even though acute caffeine intake prior to test sessions exceeded the athletes’ usual daily consumption. Likewise, no effect on performance of squat and bench presses were reported by Karayigit et al. [58], but they reported a trend (*p* = 0.057), and an 8.8% increase in squat endurance performance. The authors also reported that 10 out of 14 participants had higher values in both squat and bench press compared to the placebo, in participants who had 347 ± 56 mg/day habitual caffeine consumption [58]. Conversely, 3 mg/kg/bm of anhydrous caffeine was found to improve resistance exercise performance in a study [23] conducted with light habitual caffeine users. Habitual caffeine consumption has been suggested to increase adenosine receptor numbers and may modify cytochrome P450 enzyme function, hereby reducing the erogenicity [59]. However, Gonçalves et al., refuting and describing tolerance to caffeine as a myth, reported that performance effects of caffeine during a 30-min cycling time trial performance were not influenced by the level of habitual caffeine consumption [12]. To date, a few studies [60,61] directly investigated the moderating effect of various caffeine consumption levels (low, moderate, high) on ballistic and resistance exercise performance that reported no influence of habitual caffeine intake; however, this requires further exploration. Additionally, participants are generally asked to abstain from caffeine consumption, as the current study, for 12–24 h before test sessions in the caffeine literature. Speculation was made by Astorino et al. [13] that heavy caffeine users’ muscular endurance performance was reduced in placebo conditions compared to caffeine conditions due to the observed withdrawal symptoms, including lethargy and headaches, in which these symptoms would be ameliorated. This prior assertion was not backed by the current study, showing even low doses of caffeinated coffee increased muscular performance in caffeine naive participants, by whom possible withdrawal symptoms is not expected.

One of the novel results of the current study is the measurement, for the first time, of the effects of caffeinated coffee on muscle pain perception during resistance exercise that showed 6COF reduced pain perception during the squat exercise, but not for the bench press. Caffeine seems to increase the motivation of the participants, to complete greater repetitions to failure, by reduced pain perception, parallel to studies reporting the same effect with only higher doses [4,20]. Pain perception may be reduced with caffeine ingestion over multiple set research designs. In the current study, pain perception following three sets of squat and bench press exercises were significantly lower for the 6COF trial. Conversely, Grgic et al., suggested that 2–6 mg/kg/bm of caffeine did not reduce pain perception during a single set of squat and bench press 60% of 1 RM repetition to failure protocol [57]. Although the pain perception was reported to occur with greatest sensitivity in the luteal phase [31], effects of low doses of anhydrous or coffee form of caffeine on pain perception may be more apparent in the other phases of the menstrual cycle.

Improved pre- and post-exercise cognitive performance measured with the flanker task, as in the current study, was shown previously [28]. Similarly, Hogervorst et al., reported that 100 mg of anhydrous caffeine given three times at regular time intervals during time trial test improved cognitive performance [27]. Caffeine may therefore have an important role in individual and team sports in which concentration and reaction times have an influence on match/training performance. Previously, caffeine was shown to stimulate catecholamine secretion so increases heart rate and blood pressure during exercise, in turn, attenuating post-exercise autonomic recovery [33]. Parallelly, 400 mg of anhydrous caffeine was reported to disrupt autonomic function (LF/HF ratio) during 5- and 15-min post-exercise because of increased sympathetic nerve activity [34]. However, Sarshin et al. [35] reported both 3 and 6 mg/kg/bm anhydrous caffeine increases resting cardiac autonomic modulation and accelerates post-exercise autonomic recovery after a bout of anaerobic exercise in recreationally active young men, which was also observed in young men after a submaximal exercise test [62]. Divergent results can be related to the various factors that study designs have, such as body position, sex, age, and cardiovascular fitness [33,34,35]. In this study, 3 and 6 mg/kg/bm coffee form of caffeine administration had no effect on HRV. Speculation can be made that polyphenols have antioxidant potential, and exist in coffee, and may blunt the adverse effect of caffeine on HRV. Future research should investigate the effects of various forms of caffeine intake on HRV pre- and post-exercise in this regard.

On a practical level, it can be suggested, based on this study’s results, that caffeine naïve female athletes may benefit from 3 and 6 mg/kg/bm of caffeine provided from coffee before training or a match in the early morning to increase physical and cognitive performance. Moreover, 6 mg/kg/bm of caffeine, provided from coffee, equates to 3–4 cups, and can be preferred to decrease muscle pain perception during squat exercises rather than 3 mg/kg/bm of caffeinated coffee. Additionally, because both doses improved arousal, one may prefer to ingest 3 mg/kg/bm of caffeinated coffee in 200 or 300 mL of hot water in a more practical manner. Overall, especially those who describe themselves as “night owls”, they may ingest caffeinated coffee in the early morning, while increasing muscular and cognitive performance; they do not delay cardiac autonomic recovery, and may, in reverse, adversely affect the next training or match performance. It should be noted that brewing method, and type/brand of coffee, may alter caffeine content, and habitual daily caffeine intake of athletes may cause variable responses to the same doses of caffeinated coffee.

A few limitations inherent to our study are as follows. The effectiveness of blinding was not tested by asking participants to identify which coffee had been ingested. It is unknown whether caffeine expectancy might have affected the results of the current study. Although participants were instructed to replicate their 24-h diet prior to each test session, macronutrient intake was not analyzed. Additionally, the absence of ventilator control and blood pressure measurements before and/or during HRV may affect the results. Further, we did not measure neurotransmitter concentrations which would have provided more insights to elucidate exact mechanisms as to how both 3 and 6 mg/kg/bm of caffeinated coffee increased muscular endurance and cognitive performance, while pain perception was reduced only in 6 mg/kg/bm of caffeinated coffee conditions. Finally, the temperature of coffee was not fully standardized, as Richardson and Clarke [10] stated that the temperature at which coffee consumed can determine caffeine’s bioavailability.

## 5. Conclusions

Although upper-body muscular endurance was not affected, coffee ingestion containing 3 and 6 mg/kg/bm of caffeine increased lower body muscular endurance in the first set, and cognitive performance, without additional autonomic load. Further, 6 mg/kg/bm caffeinated coffee reduced muscle pain perception. Therefore, the current study presents evidence that low and moderate doses of caffeinated coffee provides an overall ergogenic effect on lower body muscular endurance and cognitive performance without causing an additional cardiovascular load. This may be beneficial to athletes and coaches regarding the integration of caffeine intake strategies during congested training and tournament formats.

## Figures and Tables

**Figure 1 nutrients-13-00002-f001:**
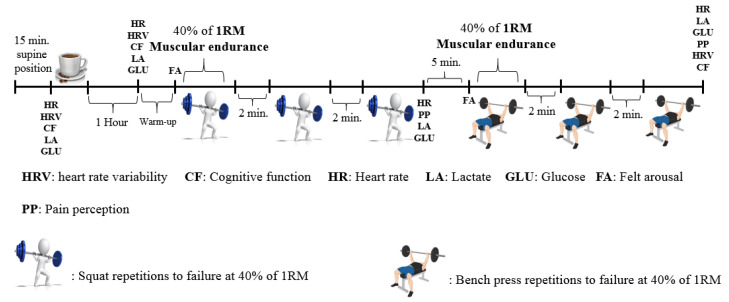
A schematic of the experimental protocol.

**Figure 2 nutrients-13-00002-f002:**
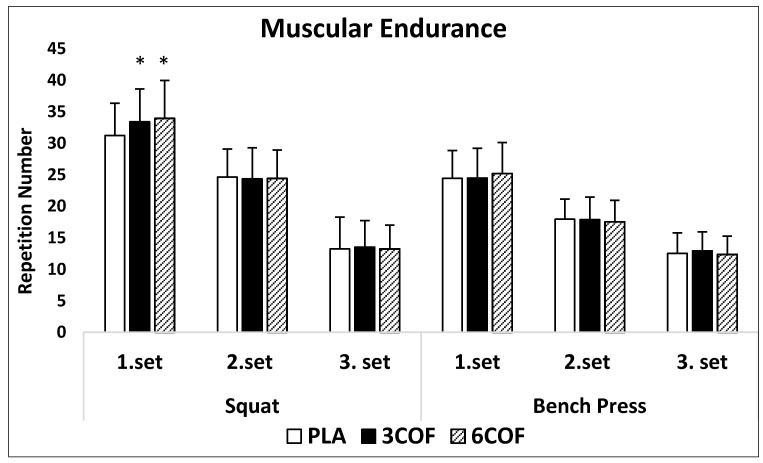
Mean (SD) repetition numbers for squat and bench press exercises over the three sets. * Significantly different from PLA.

**Table 1 nutrients-13-00002-t001:** 1 repetition maximum (RM) squat and bench press performance measured at familiarization sessions.

Variables	Test	Retest	*p*	ICC
Squat (kg)	91.2 ± 7.4 (87.3–94.9)	92.0 ± 7.1 (88.3–95.7)	0.26	0.95 (0.86–0.98)
Bench Press (kg)	63.5 ± 6.7 (60.0–66.9)	64.5 ± 8.3 (60.2–68.8)	0.23	0.94 (0.85–0.98)

Data are expressed as means ± standard deviation and (95% confidence interval (CI)). Intraclass correlation coefficients (ICC).

**Table 2 nutrients-13-00002-t002:** Heart rate variability (HRV).

	Pre Coffee			Post Coffee			Post Test		
	M	SD	95% CI	M	SD	95% CI	M	SD	95% CI
**SDNN**									
PLA	75.2	33.7	57.8–92.6	81.5	29.1	66.5–96.5	26.1	15.3	18.2–33.9
3COF	78.5	41.8	57.0–100.0	76.1	35.7	57.7–94.4	25.5	12.2	19.2–31.8
6COF	80.9	44.2	58.2–103.6	82.5	36.1	63.9–101.1	23.5	19.3	13.6–33.5
**SDSD**									
PLA	109.3	66.8	74.9–143.7	106.3	60.2	75.3–137.3	19.2	14.5	11.7–26.6
3COF	118.8	84.4	75.4–162.2	107.1	66.3	73.0–141.3	20.1	9.9	15.0–25.2
6COF	122.9	86.3	78.5–167.3	119.1	76.1	80.0–158.3	20.2	13.1	13.5–27.0
**RMSSD**									
PLA	86.4	53.1	59.0–113.7	82.9	46.9	58.8 -107.0	20.0	13.8	12.9–27.1
3COF	93.0	66.7	58.7–127.3	86.9	54.6	58.8–115.0	17.9	13.6	10.9–24.9
6COF	96.0	67.6	61.2–130.8	94.1	59.1	63.7–124.5	19.8	13.8	12.6–26.9
**TP**									
PLA	2227.9	1549.2	1931.3–3524.4	2244.4	1268.3	1592.3–2896.5	243.5	138.1	172.5–314.5
3COF	2922.7	1637.2	2080.9–3764.5	2090.4	1278.6	1433.0–2747.8	251.1	105.2	196.9–305.2
6COF	2961.5	1404.4	2239.4–3683.6	2246.2	853.6	1807.3–2685.1	278.1	189.1	180.9–375.4
**LF**									
PLA	932.7	437.0	708.0–1157.4	688.5	379.0	493.6–883.4	116.9	104.0	63.4–170.4
3COF	977.0	499.2	720.3–1233.6	739.3	464.1	500.7–977.9	124.0	77.4	84.2–163.8
6COF	1082.7	669.4	738.5–1426.9	759.8	413.8	547.1–972.6	126.6	112.5	68.8–184.4
**HF**									
PLA	1637.5	1412.6	911.2–2363.8	1125.1	765.4	731.6–1518.7	89.4	50.0	63.6–115.1
3COF	1684.9	1291.4	1020.9–2348.9	1296.1	876.9	845.2–1747.0	98.9	51.8	72.3–125.5
6COF	1547.7	894.3	1087.8–2007.5	1318.7	737.6	939.4–1697.9	82.8	65.9	48.9–116.7
**LF/HF**									
PLA	0.91	0.65	0.57–1.25	1.31	1.15	0.72–1.91	5.75	1.91	4.76–6.74
3COF	1.23	0.65	0.89–1.57	1.07	0.99	0.56–1.58	5.55	1.82	4.61–6.49
6COF	0.94	0.40	0.74–1.15	0.86	0.63	0.53–1.18	5.46	2.44	4.21–6.72
**HFnu**									
PLA	57.8	17.6	48.7–66.9	57.0	21.3	46.0–68.0	21.8	17.7	12.7–30.9
3COF	53.3	20.6	42.7–63.9	59.5	20.2	49.0–69.9	29.2	13.1	22.5–36.0
6COF	54.9	14.7	47.4–62.5	58.7	21.1	47.8–62.5	25.9	17.4	16.9–34.8
**LFnu**									
PLA	39.9	17.4	31.0–48.9	39.3	20.1	28.9–49.7	74.0	16.5	65.4–82.5
3COF	44.7	20.4	34.2–55.2	40.3	22.3	28.8–51.8	61.6	20.9	50.8–72.4
6COF	44.1	13.4	37.2–51.0	47.9	31.9	31.5–64.4	73.6	17.9	64.4–82.8

All data presented as mean ± standard deviation and 95% CI; PLA: decaffeinated coffee; 3COF: 3 mg/kg caffeinated coffee; 6COF: 6 mg/kg caffeinated coffee; SDNN: standard deviation of NN intervals; SDSD: standard deviation of successive differences; RMSSD: root mean square of successive differences; TP: total power; LF: low-frequency power; HF: high-frequency power; LF/HF: the ratio of low- and high-frequency powers; HFnu: normalized HF; LFnu: normalized.

**Table 3 nutrients-13-00002-t003:** Cognitive Performance.

	Pre Coffee			Post Coffee			Post Test		
	M	SD	95% CI	M	SD	95% CI	M	SD	95% CI
**Response Accuracy (%)—Congruent Task**									
PLA	94.1	2.7	92.7–95.5	94.2	2.9	92.7–95.7	93.4	2.3	92.2–94.6
3COF	94.5	2.4	93.2–95.7	94.2	2.7	92.8–95.6	93.7	2.6	92.3–95.0
6COF	94.5	2.5	93.2–95.8	95.2	2.4	94.0–96.5	94.1	2.8	92.6–95.6
**Response Accuracy (%)—Incongruent Task**									
PLA	92.2	2.27	91.0–93.4	92.7	2.8	91.2–94.1	92.2	2.7	90.8–93.6
3COF	92.3	2.4	91.0–93.6	92.0	2.1	90.9–93.1	93.3	2.2	92.2–94.5
6COF	91.1	2.0	90.1–92.2	91.8	2.6	90.5–93.2	93.1	2.7	91.6–94.5
**Reaction Time (ms)—Congruent Task**									
PLA	481.2	41.1	460.1–502.4	466.9	38.4	447.2–486.7	486.8	42.5	465.0–508.7
3COF	482.4	43.6	459.9–504.9	462.9	47.6	438.3–487.4	459.6	51.6	433.0–486.2
6COF	480.2	42.6	458.3–502.2	439.9	54.2	412.0–467.8	436.3	44.4	413.4–459.1
**Reaction Time (ms)—Incongruent Task**									
PLA	528.8	44.0	506.2–551.4	513.5	37.8	494.1–533.0	520.2	39.0	500.1–540.3
3COF	530.7	39.0	510.6–550.7	482.5	46.2	458.7–506.2	482.3	57.1	452.9–511.7
6COF	539.1	39.6	518.7–559.5	469.4	45.1	446.2–492.6	475.7	62.6	443.4–507.9

**Table 4 nutrients-13-00002-t004:** Lactate, Glucose, Heart Rate, Pain Perception, Felt Arousal.

	PLA			3COF			6COF		
	M	SD	95% CI	M	SD	95% CI	M	SD	95% CI
**Lactate**									
PreCof	1.16	0.22	1.05–1.28	1.19	0.19	1.09–1.29	1.30	0.21	1.19–1.41
PostCof	1.22	0.24	1.09–1.35	1.13	0.21	1.02–1.25	1.21	0.18	1.11–1.31
PostSqu	7.31	1.78	6.36–8.27	7.85	1.70	6.94–8.76	7.91	1.52	7.09–8.72
PostBch	7.89	1.65	7.01–8.77	8.77	1.97	7.72–9.82	8.98	2.17	7.82–10.14
**Glucose**									
PreCof	87.52	8.20	83.30–91.75	90.35	7.50	86.49–94.21	89.17	6.01	86.08–92.26
PostCof	90.29	9.20	85.56–95.02	91.23	9.09	86.56–95.91	83.64	7.38	79.85–87.44
PostSqu	88.58	7.90	84.52–92.65	89.47	9.80	84.42–94.51	87.58	6.71	84.13–91.04
PostBch	90.29	7.53	86.41–94.17	92.47	9.50	87.58–97.35	90.35	10.82	84.78–95.92
**Heart Rate**									
PreCof	65.94	6.84	62.42–69.45	65.52	4.96	62.97–68.08	66.23	4.69	63.82–68.65
PostCof	64.29	5.87	61.27–67.31	66.94	6.49	63.60–70.28	64.76	5.81	61.77–67.75
PostSqu	163.47	10.35	158.14–168.79	164.11	10.43	158.75- 169.48	163.58	12.19	157.32–169.85
PostBch	164.64	11.52	158.72–170.57	165.88	11.70	159.86–171.90	164.47	10.85	158.88–170.05
**Pain Perception**									
PostSqu	8.23	1.09	7.64–8.79	8.05	0.89	7.59–8.52	7.29	1.26	6.64–7.94
PostBch	6.76	2.04	5.71–7.81	6.58	2.12	5.49–7.68	6.94	1.47	6.18–7.70
**Felt Arousal**									
PreSqu	1.64	0.70	1.28–2.00	1.58	0.87	1.27–1.90	1.47	0.62	1.15–1.79
PreBch	2.52	1.06	1.98–3.07	3.47	0.62	3.02–3.92	3.64	1.11	3.07–4.22

PreCof: prior to coffee ingestion; PostCof: after coffee ingestion; PostSqu: immediately after third set of squat muscular endurance test; PostBch: immediately after third set of bench press muscular endurance test.

## Data Availability

Data is available for research purpose upon reasonable request to the corresponding author.

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
