# Peer review of "Effects of Different Doses of Caffeinated Coffee on Muscular Endurance, Cognitive Performance, and Cardiac Autonomic Modulation in Caffeine Naive Female Athletes"

_nutrients, 2020, doi:10.3390/nu13010002_

Round 1
Reviewer 1 Report
Dear authors:
First of all, I would like to thank you for preparing this research to evaluate the ergogenic properties of caffeinated coffee in a women sample. I think your work is of great interest because the majority of researches are carried out in the male population and there are differences that may affect its use in women.
Next, I will indicate some aspects that I think would improve the presented article.
- In the introduction, it is said the differences in cognitive performance during the phases of the menstrual cycle. However, no allusion has been detected to the effect of supplementation in the different stages of the menstrual cycle and its relationship with sports performance. Therefore, it is advisable to review research on this topic and assess its possible inclusion in the introduction or even to complement the discussion. Some examples of these investigations could be:
Romero-Moraleda, B .; Del Coso, J .; Gutiérrez-Hellín, J .; Lara, B. The effect of caffeine on the velocity of half-squat exercise during the menstrual cycle: a randomized controlled trial. Nutrients 2019, 11, 2662.
Lara, B .; Gutiérrez-Hellín, J .; García-Bataller, A .; Rodríguez-Fernández, P .; Romero-Moraleda, B .; Del Coso, J. Ergogenic effects of caffeine on peak aerobic cycling power during the menstrual cycle. Eur J Nutr 2020, 59, 2525–2534.
- The aim of the investigation should be clearer since it is indicated that it is the first investigation that is carried out on this topic but it does not said which is the specific objective of the research.
- In line 117, I think the closing parenthesis is missing.
- Figure 1 should appear within the Study design section instead of before it.
- In line 177, it is advisable to review the numbering of the section because it does not agree with the format of the previous and subsequent ones.
- When refers to mg/kg, it should be specified that this kg is kg of bodyweight.
- In table 1, the value indicated for Squat in Test does not correspond to that specified in the text.
- In line 466, it should be corroborated that the term “erogenicity” is correct in that context.
- In line 523 I think there is a word missing.
- I think that the conclusions should also refer to the results on upper body because it is part of the objective indicated in the abstract and it is treated throughout the text. Likewise, the differences in lower body muscular endurance are indicated globally but they only were clear in the first set.
- Reference 32 is incompleted and self citation. It would also be advisable to review the citation in the text to ensure that they are numbered in order of appearance in it and avoid all self-citations that are not necessary.
- In general, I think it would be appropriate to carry out a language review because some paragraphs are difficult to understand and some number mismatches have been detected between subject and verb.
Kind regards,
Reviewer 2 Report
This study investigated the acute ergogenic effects of coffee intake on physical and cognitive performance in women. No studies have investigated this effect in the female groups and therefore fills a gap in the literature. Despite the fact that the study design seems to be solid, there are several inconsistencies, which have to be explained or changed. My suggestions are outlined below.
Line 27-29: Probably it is not 3 mg/kg of caffeinated coffee (3 COF) but 3 mg/kg of caffeine provided from coffee
Line 31: Please, explain the abbreviations
Introduction:
The introduction could more focus on why authors believe responses would be different between males and females. That would provide better support for why this study is needed.
57:Please, provide the reference
Materials and Methods
117: Please, explain the abbreviation (RM)
115: Characteristic of participants is not the same as in abstract (23.6 ± 2.7 years, body mass = 64.2 ± 4.5 kg, height= 168 ± 3 cm)
134: (d) “naïve caffeine consumer” [ref.] instead of “daily caffeine intake ≤ 135 25 mg/day….”.
Information in lines 135-136 is similar as in lines 130-131 and it is not necessary
Figure 1.: The figure is not on an appropriate level- modify it (especially red underlines)
148: as in abstract in lines 27-29; Probably it is not 3 mg/kg of caffeinated coffee (3 COF) but 3 mg/kg of caffeine provided from coffee
162: standardized static-dynamic stretching- what kind of exercises?
160: Please delete a dot after 500.
173: How did they report it? What was the mean total energy intake? Did you notice differences between trials?
176: How did you control it? Why did you choose this phase? Could it affect on results of pain perception/ exercise performance?
RM and Muscular Endurance Test Protocol
179: please change the order of “%” and numbers
186- 187 – what temp was used for the strength-endurance test? If controlled (like in 1RM test 2/0/2/0) when was the point of failure - when the participant did not complete the rep or when the participant did not follow the tempo. If was maximal movement tempo during the strength-endurance test you can use according to PMID: 30777094
191: What was the grip width during the bench press exercise? It should be specified PMID: 31531132
Line 193-195 – Why tempo was 2 ecc and 2 con second? According to the results of studies PMID: 32735429; PMID: 32390725; PMID: 32269656 the 1RM test should be performed at that same tempo as experimental trials.
If not please add such information to the limitation of the study.
Statistical Analysis&Results
The statistical section, as well as the results section, is confusing. Please make it clear for which data two-way anova was conducted? and for which one-way? I think that you are mistaking two-way with one-way anova several times in the results section, please check this section carefully.
Please provide effect size ranges in statistical section.
Line 247: if you assess the normality of the data you should report that in the results section.
364: in ref. [15] they did not notice effects on muscular endurance,
412: Please provide a reference
471: It was analyzed: check https://pubmed.ncbi.nlm.nih.gov/30702372/, https://pubmed.ncbi.nlm.nih.gov/32859145/
505-51: as in abstract/ lines 148
please delete dots after Mg. and Gr. in all manuscript
Round 2
Reviewer 2 Report
The reviewer would like to thank the authors for considering the comments and suggestions.